# Fungal β-glucan-facilitated cross-feeding activities between Bacteroides and Bifidobacterium species

Pedro Fernandez-Julia[1], Gary W. Black[1], William Cheung[1], Douwe Van Sinderen [2] & Jose Munoz-Munoz [1✉]

The human gut microbiota (HGM) is comprised of a very complex network of microorganisms, which interact with the host thereby impacting on host health and well-being. β-glucan has been established as a dietary polysaccharide supporting growth of particular gut-associated bacteria, including members of the genera *Bacteroides* and *Bifidobacterium*, the latter considered to represent beneficial or probiotic bacteria. However, the exact mechanism underpinning β-glucan metabolism by gut commensals is not fully understood. We show that mycoprotein represents an excellent source for β-glucan, which is consumed by certain *Bacteroides* species as primary degraders, such as *Bacteroides cellulosilyticus* WH2. The latter bacterium employs two extracellular, endo-acting enzymes, belonging to glycoside hydrolase families 30 and 157, to degrade mycoprotein-derived β-glucan, thereby releasing oligosaccharides into the growth medium. These released oligosaccharides can in turn be utilized by other gut microbes, such as *Bifidobacterium* and *Lactiplantibacillus*, which thus act as secondary degraders. We used a cross-feeding approach to track how both species are able to grow in co-culture.

[1] Microbial Enzymology Lab, Department of Applied Sciences, Northumbria University, Newcastle Upon Tyne NE1 8STTyne & Wear, EnglandUK. [2] School of Microbiology & APC Microbiome Ireland, University College Cork, Cork, Ireland. ✉email: jose.munoz@northumbria.ac.uk

The human gut microbiota (HGM) is a complex ecosystem of microbes, which are purported to beneficially impact on human health. Dietary carbohydrates represent the main energy source for the human body, though we lack the enzymatic capabilities to degrade most of these glycans ourselves[1]. However, the HGM encodes an arsenal of carbohydrate active enzymes (CAZYmes) able to catabolize such carbohydrates[1]. The metabolic end products of glycan fermentation by the HGM are mostly SCFAs, such as propionate, acetate and butyrate, which have a variety of beneficial effects on the human host. Imbalance in the composition of the gut microbial community, sometimes referred to as dysbiosis, is a characteristic of Inflammatory Bowel Disease (IBD), colorectal cancer, obesity, *Clostridioides difficile* infections and, potentially, a wide range of other conditions[2–4].

As stated above, the HGM is a complex network of microbes representing approximately 10 trillion bacteria[5]. Nonetheless, just two phyla are dominant in this intricate bacterial community, i.e. Bacteroidota and Bacillota, complemented with representatives of several minor phyla, such as Actinomycetota or Verrucomicrobiota. Various members of the *Bacteroides* genus have been shown to contain specific clusters of co-regulated genes which metabolise a specific glycan, and which are called Polysaccharide Utilisation Loci (singular: PUL, plural: PULs). A given PUL contains the genes needed to sense, transport and degrade a particular glycan[6–8]. For example, the *Bacteroides thetaiotaomicron* (*Ba. thetaiotaomicron* VPI-5482, BT) genome harbours 96 different PULs, according to the CAZY database[9,10], while another *Bacteroides* species, *Ba. ovatus* ATCC8483 (Bacova), encompasses 115 different PULs. Generally, a cell surface-associated glycoside hydrolase (GH) or polysaccharide lyase (PL) initiates extracellular polysaccharide degradation allowing the release of the resulting oligosaccharides into the growth medium[11–17].

Members of the genus *Bifidobacterium* are particularly abundant in full-term, breast-fed infants, where they are believed to exert important beneficial effects[18–21]. The dominance of these bacteria in this niche is attributed, at least in part, to their ability to metabolise (particular) human milk oligosaccharides (HMOs) as a sole carbon and energy source[22]. Different bifidobacterial species have particular HMO consumption preferences, for example strains of *Bi. breve* and *Bi. longum* are known to internalise and metabolise Lacto-N-tetraose[23–26], while *Bi. catenulatum* subsp. *kashiwanohense* can use 2-fucosyllactose as a carbon source[19,27]. The prototypical strain *Bi. breve* UCC2003 has been shown to metabolise several HMOs and other dietary poly/oligosaccharides either on its own[23,28] or through cross-feeding involving other members of the gut microbiota; examples of such saccharidic substrates are mucin[29,30], arabinogalactan/galactan[31,32] and sialyllactose[20].

β-glucan is a complex glycan which has been explored as a potential prebiotic and which is particularly abundant in cereals and fungal cell walls[13,33–36]. Cereal-derived β-glucan, with a defined β-1,3/1,4-mixed linkage (Fig. 1A), has been employed to understand the molecular mechanisms by which certain *Bacteroides* species are able to sense, internalize and degrade this polymer[13,34–36]. For example, Bacova employs a PUL with a cell surface-associated glycoside hydrolase 16 (GH16) and a periplasmic GH3 to catalyse degradation of this mixed linkage cereal β-glucan[34–36].

Furthermore, a *Ba. uniformis* ATCC8492 PUL, which is involved in the metabolism of related β-1,3-glucans present in yeast and the seaweed glucan laminarin (Fig. 1A), is responsible for the initial degradation at the cell surface by a glycoside hydrolase 16 (GH16) and a GH158, and a subsequently acting periplasmic GH3[13]. In addition, *Ba. uniformis* strain JCM13288 can degrade laminarin, pustulan and porphyran β-glucans, employing two PULs encoding a GH16, GH30, GH158 and GH3,

and sharing oligosaccharides with other gut microbiota members, possibly supporting gut homeostasis[37].

Another type of β-1,6-glucan, which is found as linear glucan in the lichen *Lasallia pustulata* and as linear glucan but cross-linked with other cell wall components in the yeast *Saccharomyces cerevisiae* or in the almond mushroom (*Agaricus blazei*), is used by BT through a PUL involving just a surface-located GH30_3 and a periplasmic GH3[17,33]. Little is known about the degradation of fungal β-glucan, such as that derived from mycoprotein produced by *Fusarium venenatum*, a fungus employed by Quorn® as a functional food ingredient[38]. The chemical structure of this β-glucan consists of a linear β-1,3-glucan backbone carrying side chains of β-1,6-glucans, which makes it distinct from cereal β-glucan, or yeast β-glucan/laminarin, where either the core chain or the side chain linkages are different, respectively[13,34,39] (Fig. 1A).

Currently, there are only a few publications describing any health benefits elicited by Bacteroidota (in fact some species are considered opportunistic pathogens)[40,41]. However, a case can be made that certain *Bacteroides* species represent primary glycan degraders that allow carbohydrate cross-feeding by other, beneficial members of the microbiota, such as bifidobacteria (which are not known to be able to metabolise β-glucan directly). Such metabolic interactions occur between BT and *Bi. longum*, and between *Ba. cellulosilyticus* DSMZ14838 (Baccell DSMZ) and *Bi. breve* UCC2003 when grown on larch arabinogalactan protein (AGP) as the sole carbon source[31,42,43]. Other trophic interactions have been established between various species of *Bacteroides* and *Bifidobacterium*[39,44–48], but have yet to be defined for mycoprotein β-glucan.

In the current study, we performed a molecular characterization of mycoprotein and yeast β-glucan degradation by Bacova, *Ba. cellulosilyticus WH2* (Baccell WH2) and BT, highlighting the PUL architecture of these *Bacteroides spp.* In addition, we uncovered cross-feeding interactions between *Bacteroides*, as the primary degrader, and certain species of *Bifidobacterium* and *Lactiplantibacillus*, acting as secondary degraders of Mycoprotein-derived fungal β-glucan.

## Results

**Screening *Bacteroides* species for β-glucan metabolism.** Certain members of the *Bacteroides* genus have been established as generalist fermenters of mixed linkage (barley) and yeast β-glucan[13,34–37]. The genomes of these *Bacteroides* species encompass various PULs (here referred to as glucan utilization loci or GULs) to degrade this complex glycan, employing encoded glycoside hydrolases for this metabolic process. To assess the ability of *Bacteroides* species to use fungal-derived β-glucan, we extracted this polysaccharide from mycoprotein of *Fusarium venenatum* (see "Materials and methods") and screened the 23 most prominent, gut-associated Bacteroidetes species for their ability to grow on this glycan as well as on β-glucan derived from yeast (Table S1). Under the (anaerobic) conditions used, *Ba. xylanisolvens*, *Ba. intestinalis*, Bacova, *Ba. fragilis*, *Ba. finegoldii*, *Ba. vulgatus*, *Ba. uniformis*, BT (partial growth), *Dysgonomonas gadei*, *D. mossii* and two strains of Baccell (Baccell WH2 and Baccell DSMZ) were shown to metabolise β-glucan derived from mycoprotein. In addition, *Ba. vulgatus*, *Ba. uniformis*, Bacova, BT, *Dysgonomonas gadei*, *D. mossii* and both Baccell strains were able to utilise yeast β-glucan. Bacova has previously been shown to grow in barley-derived β-glucan as well[34,35], showing this species' broad ability to ferment β-glucans from different sources and chemical structures (Fig. 1A). Figure 1B shows the growth profile of Baccell WH2, Baccell DSMZ, BT and Bacova on β-glucan from mycoprotein (Fig. 1B). In addition to these growth experiments, we assessed growth of various *Bifidobacterium* strains on

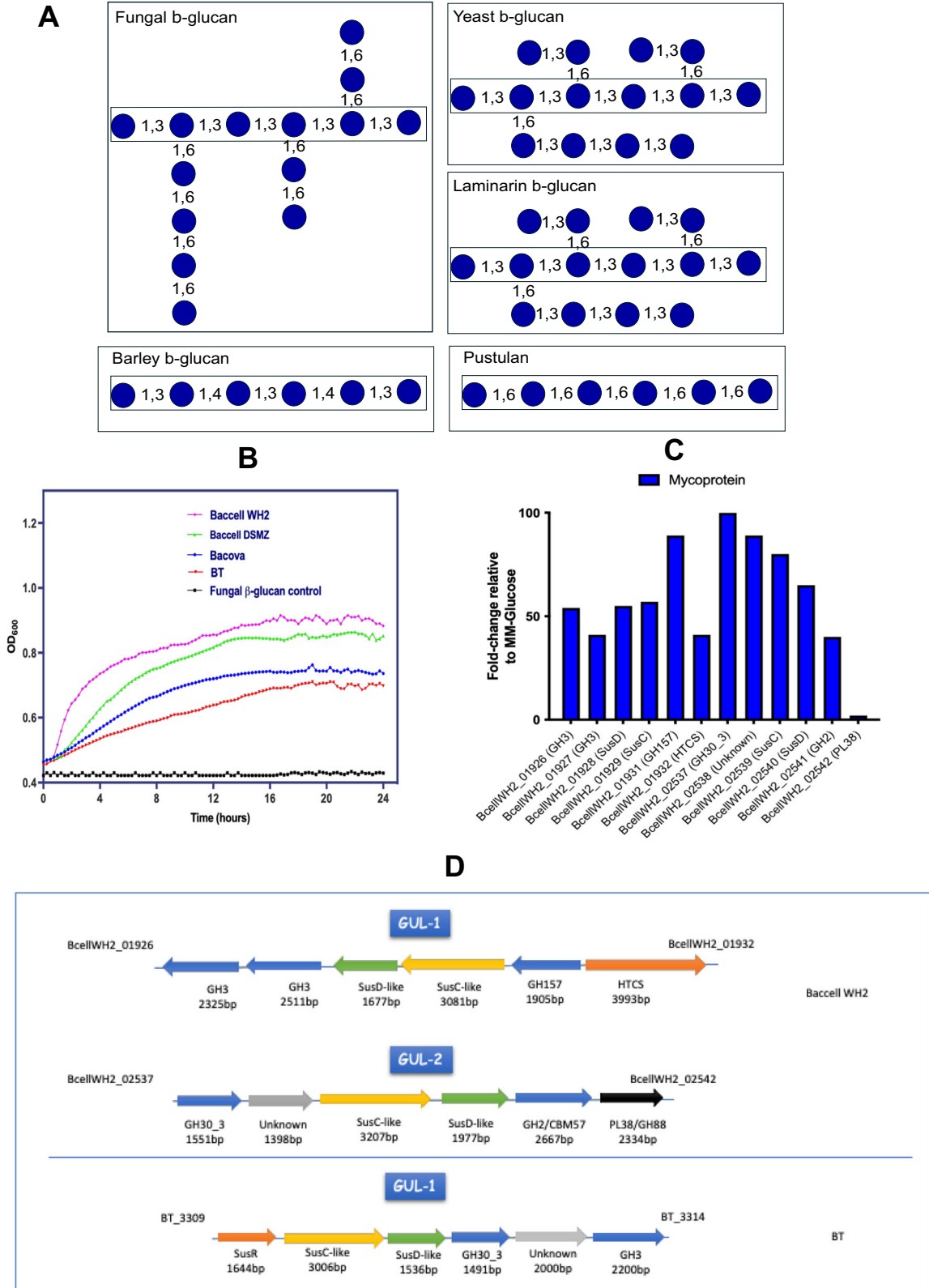

**Fig. 1 Polysaccharide utilization systems in Bacteroides species for β-glucan degradation. A** Structures of fungal (mycoprotein), yeast, laminarin, barley and pustulan β-glucan. **B** Growth *Bacteroides ovatus*, DSM, WH2 and BT on mycoprotein as measured on a plate reader. All growths have been produced in 3 different independent replicates (n = 3). **C** Number of proteins expressed by Baccell WH2 on the use of β-glucan from Mycoprotein as identified by proteome analysis. The proteomics have been produced in 3 different independent replicates (n = 3). **D** GUL structure in Baccell WH2 (GUL-1 and GUL-2) and BT acting on mycoprotein, yeast and pustulan.

mycoprotein-derived β-glucan, clearly showing a lack of ability by these strains to use this complex carbon source (Fig. S1A).

**GUL regulation in *Bacteroides* and architecture**. To further characterise growth of selected *Bacteroides* strains on particular β-glucans, we performed proteomics analysis on Baccell WH2 when grown on glucose (acting as reference) or mycoprotein-derived fungal β-glucan as carbon sources. We compared the generated proteome data of this bacterium when grown on either of these carbon sources to identify proteins that exhibit increased expression when the strain is grown on the complex polymer. As shown in Fig. 1C, this analysis revealed that all proteins encoded by two assigned GULs (named here as GUL-1 and GUL-2 and representing locus tags BcellWH2_01929-BcellWH2_01932 and BcellWH2_02537-BcellWH2_02542, respectively, see Fig. 1D) exhibit increased expression when Baccell WH2 was grown on mycoprotein-derived fungal β-glucan metabolism (when compared to growth on glucose). GUL-1 was predicted to encode two GH3 enzymes (BcellWH2_01926 and BcellWH2_01927) and a GH157 member (BcellWH2_01931), while the proteins encoded by locus tags BcellWH2_01928 and BcellWH2_01929 represent the SusC/D-like pair predicted to be involved in (polysaccharide) substrate binding and recognition at the bacterial cell surface (Fig. 1D). Furthermore, BcellWH2_01932 is predicted to represent the Hybrid Two Component System (HTCS) sensing system controlling transcriptional regulation of GUL-1. GUL-2 encodes a GH30_3 (BcellWH2_02537, a predicted endo-β-1,6-glucanase according to the CAZY database), a GH2 (BcellWH2_02541) and a protein without known function (BcellWH2_02538). In addition to these proteins, BcellWH2_02539 and BcellWH2_02540 represent the predicted the SusC/D pair in GUL2 (Fig. 1D). To confirm the proteomics data, we grew Baccell WH2 to mid-exponential and performed RT-qPCR on selected SusC/D pairs identified in the above described GULs (Fig. S1B). In addition, we also used this approach with BT and Bacova on the SusC/D pairs identified based on differential expression patterns when these two strains had been grown on beta-glucan from pustulan[17,34,35] (Fig. S1B). Since we observed differential expression based on RT-qPCR, it seems that their corresponding GULs, which are substantially different from that of Baccell WH2 in terms of their genetic structure and content, are also responsible for growth on fungal β-glucan. Figure 1D and S1C show the GUL architecture of BT, Bacova and Baccell WH2, and predicted functions pertaining to GUL regulation, β-glucan degradation and associated oligosaccharide intake. BT and Baccell WH2 employ a different GUL architecture for fungal β-glucan utilization (Fig. 1D), when compared to Bacova, which based on its expression profile appears to use the same GUL for fungal and barley β-glucan Figs. S1B and S1C.

As stated above, BT, Baccell WH2 and Bacova can metabolise fungal β-glucan. Based on qPCR performed on BT growing in a medium containing either fungal β-glucan or pustulan, this bacterium appears to employ the same GUL for either of these two carbon sources[17] (BT3309-BT3314, Fig. 1D and S1B). According to qPCR data obtained when Bacova is growing in a medium supplemented with either barley or fungal β-glucan (Fig. S1B), the bacterium was shown to employ the same GUL for either of these carbon sources, which contrasts with what we discovered for Baccell WH2, as this bacterium appears to employ distinct GULs to deconstruct either fungal or barley β-glucan (Fig. 1D and S1C)[49].

**Dissecting the enzymes encoded by GUL-1 and GUL-2 that degrade fungal β-glucan**. Following our observation that certain proteins encoded by, and their corresponding genes present in,

GUL-1 and GUL-2 of Baccell WH2 elicit increased expression when growing on fungal β-glucan, we wanted to confirm their involvement in this metabolic process. For this purpose, we recombinantly expressed the enzymes representing the putative GH157, GH30_3 and GH3 (Baccell WH2_01926) activities from GUL1/GUL2 to fully dissect the mechanism of degradation of this complex dietary carbon source. For this purpose, we incubated the expressed proteins with fungal β-glucan and revealed possible degradation products by means of HPLC chromatography. More specifically, Fig. 2A shows the HPLC chromatograms of BcellWH2_01931 and BcellWH2_02537 (representing predicted GH157 and GH30_3 activities, respectively) acting on fungal β-glucan.

The data presented in Fig. 2A and 2B confirmed that BcellWH2_01931 acts on fungal β-glucan (Fig. 2A) and the linear β-1,3-glucan from *Euglena glacialis* (Fig. 2B) initially releasing penta- and trisaccharides, which are converted to disaccharides and glucose upon longer incubation (16 h). However, we did not find any activity when the enzyme was incubated with pustulan. Table 1 lists the kinetic parameters ($k_{cat}/K_m$) of this protein acting on either substrate.

To validate the prediction that these enzymes are involved in fungal β-glucan metabolism, we performed protein location analysis by LipoP[50], indicating that BcellWH2_01931 (GH157) and BcellWH2_02537 (GH30_3) are located at the cell surface of Baccell WH2. Furthermore, according to the CAZY database, the GH157 enzyme was anticipated to act as an endo-β-1,3-glucanase, while the GH30_3 enzyme was expected to possess endo-β-1,6-glucanase activity as has been reported for other members of this CAZY family[17]. Like GH157, recombinantly expressed and purified GH30_3 was incubated with fungal β-glucan, linear β-1,3-glucan and pustulan (linear β-1,6-glucan), followed by HPLC analysis. Figure 2C and 2D show the associated chromatography results for this incubation experiment, confirming the β-1,6-glucanase activity with fungal β-glucan (Fig. 2C) and pustulan (Fig. 2D). In addition, GH30_3 didn't show any activity against linear β-1,3-glucan. Table 1 displays the catalytic parameters of this protein as measured by DNSA assays with fungal β-glucan and pustulan. BcellWH2_02537 showed activity parameters for β-glucan from *Fusarium venenatum* that are like those obtained for pustulan ($k_{cat}/K_m$ of $2512 \pm 28$ and $1968 \pm 17$ mg ml$^{-1}$ min$^{-1}$) suggesting that the enzyme does not require interactions with the β-(1,3)-glucan backbone. In addition, BcellWH2_02537 was shown to only exhibit activity on oligosaccharides larger than β-(1,6)-glucotriose indicating that, as in BT3312, the enzyme has 3 subsites in the active site (nomenclature established by Davies et al.)[51].

Finally, to fully understand how mycoprotein-derived β-glucan is metabolised by Baccell WH2, we recombinantly expressed the predicted GH3 and GH2 enzymes (as specified by BcellWH2_01926 and BcellWH2_02541, respectively), showing that both are able to act on the different oligosaccharides produced by the action of the GH157 and GH30_3 enzymes. BcellWH2_01926 (GH3) was able to degrade β-1,6-glucooligosaccharides released by GH30_3, while the GH2 enzyme (BcellWH2_02541) was shown to act on β-1,3-glucooligosaccharides (glucobiose and glucotriose), in both cases releasing glucose as the final product, which confirms the exo-acting mechanism for these enzymes (BcellWH2_01926 and BcellWH2_02541). Table 1 also displays the catalytic parameters of these two exo-glucosidases, which highlights that the activity of the GH2 enzyme is similar for β-1,3-glucobiose and β-1,3-glucotriose, suggesting 2 subsites in the active site.

In the case of BT, the GUL architecture is simpler than that observed for Baccell WH2 (Fig. 1D). For the former bacterium, only the GH30_3 (BT3312) and a GH3 (BT3314) have previously been described to be active on pustulan β-glucan[17]. To assess the

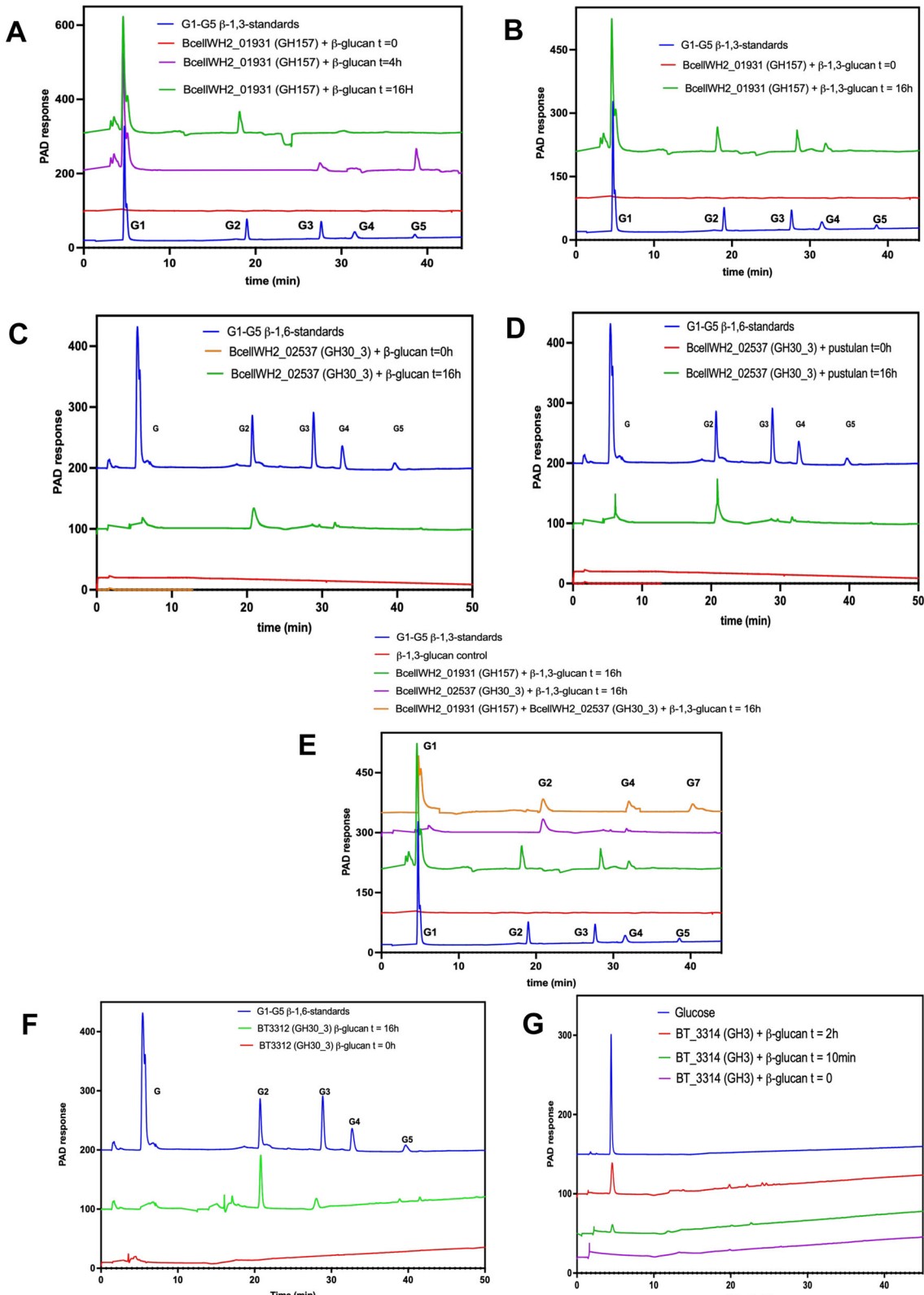

**Fig. 2 HPLC analysis of the enzymatic reactions of glycoside hydrolases in Baccell WH2 and BT on fungal β-glucan.** All HPLC experiments have been produced in 3 different independent replicates (n = 3). **A** Time course of BcellWH2_01931 (GH157) with mycoprotein β-glucan. **B** Time course of BcellWH2_01931 (GH157) with linear β-1,3-glucan. **C** Time course of BcellWH2_02537 (GH30_3) on mycoprotein β-glucan. **D** Time course of BcellWH2_02537 (GH30_3) with pustulan. **E** Time course of BcellWH2_01931 (GH157) and BcellWH2_02537 (GH30_3) together on linear β-1,3-glucan. **F** HPLC of BT3312 (GH30_3) on mycoprotein β-glucan. **G** HPLC of BT_3314 (GH3) on mycoprotein β-glucan.

**Table 1 Kinetic parameters of GHs in Baccell WH2, BT and *Bi. breve* UCC2003 acting different β-glucan substrates and gluco-oligosaccharides.**

| | Substrate | $K_{cat}/K_m$ (mg/ml/min) | $k_{cat}/K_m$ (µM/min) |
|---|---|---|---|
| **Baccell WH2** | | | |
| BcellWH2_01926 (GH3) | β-(1,6)glucobiose | | 10.2 ± 0.9 |
| | β-(1,6)glucotriose | | 25.2 ± 1.8 |
| BcellWH2_01931 (GH157) | *Fusarium* β-glucan | 1069 ± 11 | |
| | Linear β-(1,3)glucan | 2611 ± 38 | |
| | Pustulan | Not detectable | |
| BcellWH2_02537 (GH30_3) | *Fusarium* β-glucan | 2512 ± 28 | |
| | Pustulan | 1968 ± 17 | |
| | Linear β-(1,3)glucan | Not detectable | |
| | β-(1,6)glucotriose | | 2.5 ± 0.1 |
| | β-(1,6)glucohexaose | | 9.8 ± 0.9 |
| BcellWH2_02541 (GH2) | β-(1,3)glucobiose | | 18.1 ± 1.2 |
| | β-(1,3)glucotriose | | 35.1 ± 2.9 |
| **BT** | | | |
| BT3312 (GH30_3) | *Fusarium* β-glucan | 2874 ± 35 | |
| | Pustulan[a] | 1776 ± 20 | |
| | Linear β-(1,3)glucan | Not detectable | |
| | β-(1,6)glucotriose[a] | | 1.7 ± 0.1 |
| | β-(1,6)glucohexaose[a] | | 6.0 ± 0.6 |
| BT3314 (GH3) | β-(1,6)glucobiose[a] | | 5.6 ± 0.2 |
| | β-(1,6)glucotriose[a] | | 6.7 ± 0.6 |
| ***Bi. breve* UCC2003** | | | |
| Bbr_0109 (GH1) | β-(1,3)glucobiose | | 12.7 ± 1.8 |
| | β-(1,4)glucobiose | | 8.1 ± 0.9 |
| | β-(1,6)glucobiose | | 5.4 ± 0.6 |

[a]Data taken from Temple et al. as comparison[17].

catalytic activity of BT3312 on β-glucan from *Fusarium venenatum*, we performed enzymatic assays with this polysaccharide obtaining activity levels that are similar to those of BcellWH2_02537 ($2874 ± 35$ mg ml$^{-1}$ min$^{-1}$ for BT3312 and $2512 ± 28$ mg ml$^{-1}$ min$^{-1}$ for BcellWH2_02537) (Table 1). The BT3312 enzyme was shown to be active on fungal β-glucan side chains, thereby releasing particular β-1,6-oligosaccharides, which in turn can be hydrolysed by BT3314 as an exo-glucosidase to release glucose (Fig. 2F for BT3312, and Fig. 2G for BT3314). In the same way as Baccell WH2 enzymes, Table 1 shows the catalytic parameters for BT3312 and BT3314 on these substrates.

**Structural modelling of GH157 and GH30_3 from Baccell WH2.** To dissect the interactions of the Baccell WH2-encoded GH157 and GH30_3 enzymes with fungal β-glucan, we attempted to obtain crystals of the Baccell WH2 proteins. Unfortunately, despite screening multiple conditions we were unable to obtain suitable crystals. Instead, we obtained the structure of the GH30_3 protein by comparison to the crystal structure solved for BT3312 using the Phyre2 algorithm as a search platform[17,52]. Fig. S1D shows that the predicted structure is an (β/a)$_8$ barrel with the conserved retaining mechanism where two glutamic acids act as nucleophile (E339 and E347 for BT3312 and BcellWH2_02537, respectively) and acid/base (E238 and E247 for BT3312 and BcellWH2_02537, respectively). As indicated in this Figure, both proteins show a high level of sequence similarity (58.33% identical) and exhibit the same activity profile (Table 1). This indicates that the GH30_3 from Baccell WH2 contains only three major subsites in a similar manner to what has been described for BT3312[17]. Amino acids involved in the binding and catalysis are conserved in both proteins (Fig. S1E).

**Oligosaccharides released into the cultivation medium.** To investigate if Baccell WH2 and BT release oligosaccharides into

their cultivation medium when grown on fungal β-glucan, thereby perhaps allowing growth of other bacteria through cross-feeding activities, we obtained cell free growth medium following cultivation of these strains to the mid-exponential and stationary phase (Fig. 3). The presence of oligosaccharides released into the media by Baccell WH2 or BT was then assessed by HPLC (Fig. 3A, B). When these two bacterial species use mycoprotein as their sole carbon source (Fig. 3A), they were shown to release oligosaccharides into the media, but these were different in each case (Fig. 3B), which is consistent with their different GUL architecture and associated enzymes, thus confirming the distinct degradative strategy followed by each of these two species.

To investigate the nature of these oligosaccharides, we performed LC/MS to determine their mass. Figure 3C shows the HPLC profile of the purified main oligosaccharide released by Baccell WH2 and Fig. 3E the associated mass spectrum of this oligosaccharide confirming that this bacterium predominantly releases a heptasaccharide. These results are in concordance with the in vitro enzymatic digestion of fungal β-glucan because when we incubated this fungal carbon source with GH30_3 and GH157 together, the products generated by both enzymes were glucose, 1,6-β-glucobiose 1,3-β-glucotetraose and the heptasaccharide present in the growth medium generated by Baccell WH2 (Fig. 2E). Figure 3F displays the mass spectra of the main oligosaccharides released by BT in the growth medium, confirming the presence of a disaccharide as the main product, corresponding to 1,6-β-glucobiose as was indicated in the HPLC chromatogram (Fig. 3D).

**Cross-feeding Bacteroides/Bifidobacterium/Lactiplantibacillus.** After we confirmed the release of oligosaccharides into the growth medium by *Bacteroides* when grown on mycoprotein β-glucan, we hypothesised that this phenomenon would allow other gut commensals to cross-feed on such released oligosaccharides.

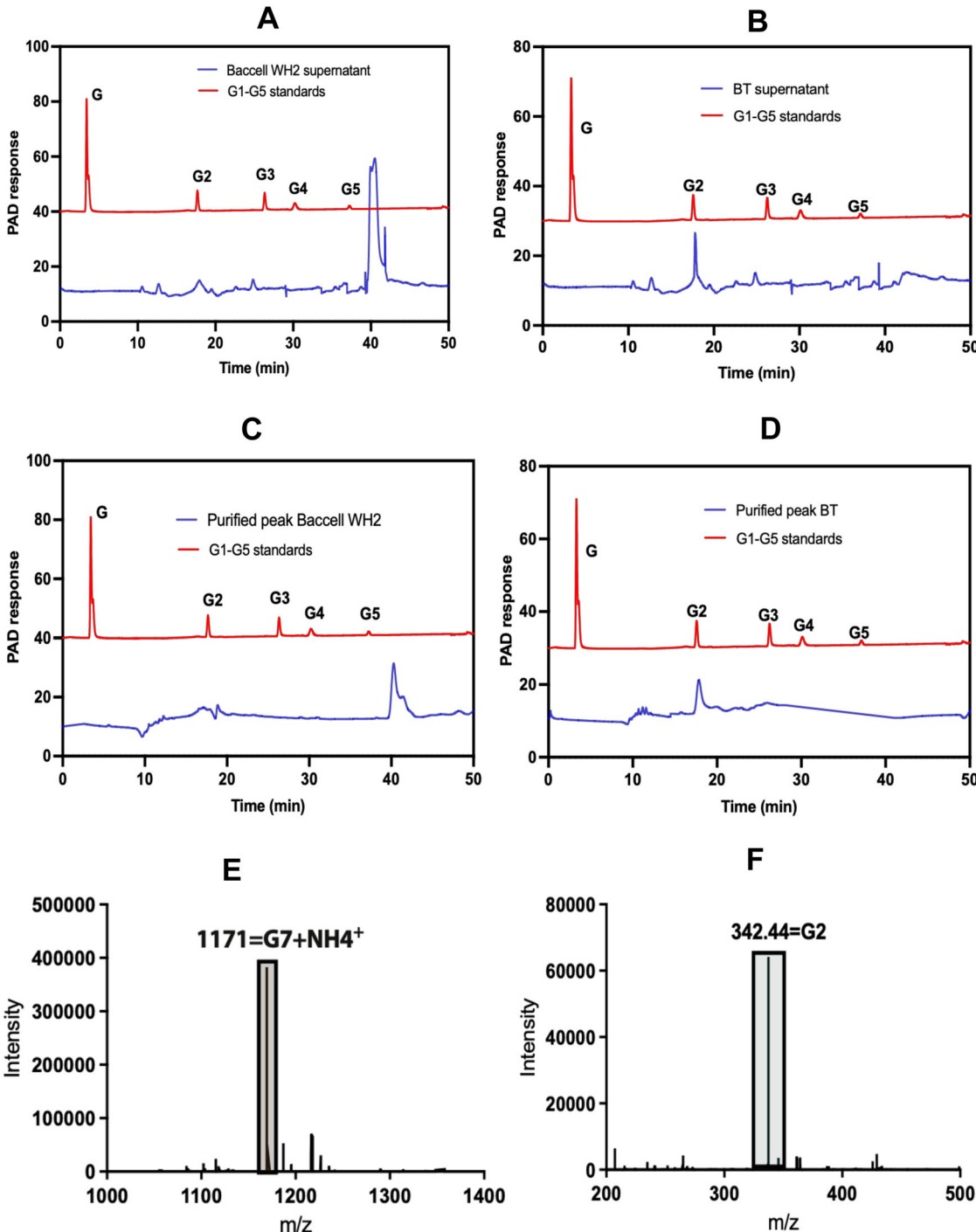

**Fig. 3 Characterization of oligosaccharides released by Bacteroides when using β-1,3-glucan. A** HPLC chromatogram of the growth media of Baccell WH2 on fungal β-glucan. **B** Same as Panel A with BT. **C** Purified oligosaccharide from Baccell WH2 after Gel Filtration (GF) column. **D** Purified oligosaccharide from BT after GF column. **E** LC/MS of Baccell WH2 supernatant grown on fungal β-glucan. **F** LC/MS of BT supernatant grown on fungal β-glucan. All HPLC and LC/MS experiments have been performed in 3 different independent replicates ($n = 3$).

Indeed, Fig. 4 shows that *Bifidobacterium strains* are able to grow when co-cultivated with *Bacteroides* on fungal β-glucan. We screened the overnight supernatant from Baccell WH2 and BT grown in β-glucan with several available *Bifidobacterium* and *Lactobacillus* strains showing that, in the case of Baccell WH2 supernatant, *Bi. breve* UCC2003, *Bi. longum* subsp. *longum* NCIMB 8809 and *Lb. plantarum* WCFS1 are unable to use intact β-glucan as a carbon source, yet that they can utilise the heptasaccharide released by Baccell WH2 (Fig. 4A). We confirmed

the ability of these strains to use the heptasaccharide testing the growth media before and after the *Bi. breve* UCC2003, *Bi. longum* subsp. *longum* NCIMB 8809 and *Lb. plantarum* WCFS1 growth. Figure 4B confirms the use of the heptasaccharide by *Bi. longum* subsp. *longum* NCIMB 8809, with *Bi. breve* UCC2003 and *Lb. plantarum* WCFS1 exhibiting a more modest ability to use the released oligosaccharide too. After 24 h fermentation, *Bi. longum* subsp. *longum* NCIMB 8809 reached a final optical density of 0.8 when using the supernatant from Baccell WH2. *Bi. breve* and *Lb.*

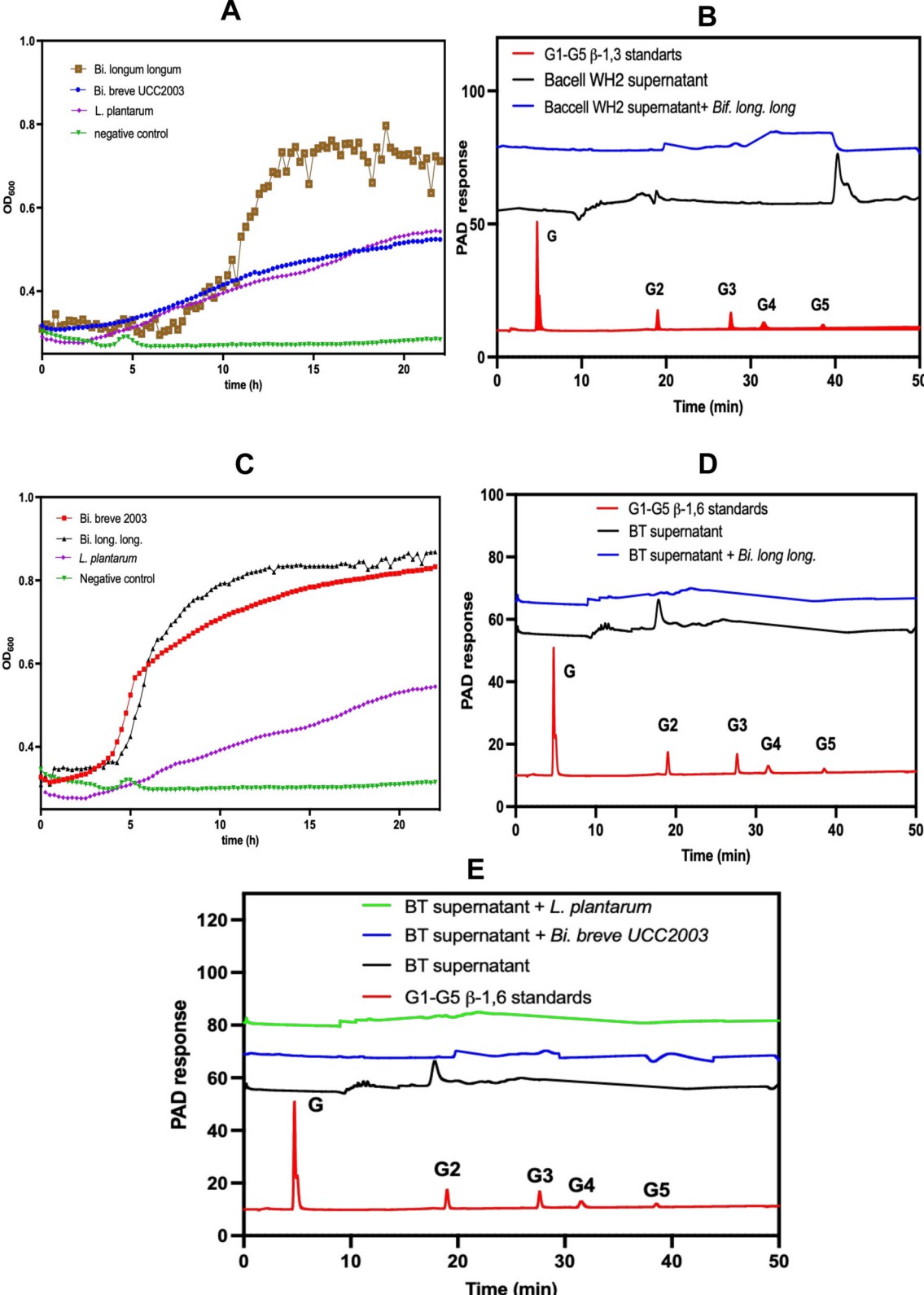

**Fig. 4 Consumption of β-glucan oligosaccharides released by *Bacteroides* into the growth media by Bifidobacterium species. A** Growth of *Bifidobacterium* with fungal β-glucan supernatant from Baccell WH2. **B** HPLC analysis of supernatants before and after growth of *Bifidobacterium longum subsp. longum* on supernantants from Baccell WH2. **C** Growth of *Bifidobacterium* with fungal β-glucan supernatant from BT. **D** HPLC analysis of supernatants before and after growth of *Bifidobacterium* longum *subspecie longum* on cell-free supernatant of BT grown on fungal β-glucan. **E** HPLC analysis of supernatants before and after growth of *Bi breve UCC2003* and *L. plantarum* on cell-free supernatants of BT grown on fungal β-glucan. All growths and HPLC experiments have been produced in 3 different independent replicates ($n = 3$).

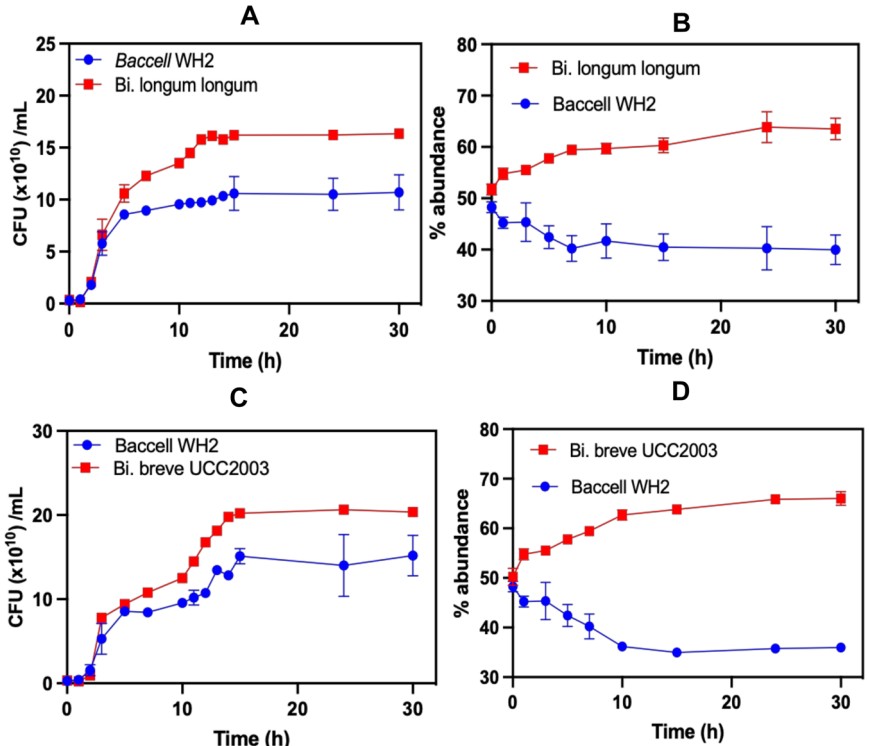

**Fig. 5 Cross-feeding experiments between Baccell WH2 and *Bifidobacterium* and *Lactiplantibacillus spp*. A** Colony forming units of Baccell WH2 + *Bi. longum subsp. longum*. **B** Percentage of Baccell WH2 + *Bi. longum subsp. longum*. **C** Colony forming units of Baccell WH2 + *Bi. Breve* UCC2003. **D** Percentage of Baccell WH2 + *Bi. breve* UCC2003. All cross-feeding experiments have been produced in 3 different independent replicates ($n = 3$).

*plantarum* were able to use these oligosaccharides but to a lesser degree, reaching a final optical density of 0.6.

When the supernatant of BT was used as carbon source for the screening of *Bi. breve* UCC2003, *Bi. longum* subsp. *longum* NCIMB 8809 and *Lb. plantarum* WCFS1, the disaccharide which is otherwise accumulating in the medium is now not present, indicating that this β-1,6-glucobiose is used by the bifidobacterial and *Lactiplantibacillus* strains to sustain their growth (Fig. 4C, D). Analysis of the growth medium by HPLC was also performed for *Bi. breve* UCC2003 and *Lb. plantarum* WCFS1 confirming their ability to use the β-1,6-glucobiose as carbon source (Fig. 4E).

To confirm this newly discovered cross-feeding interaction between *Bacteroides* and *Bifidobacterium* species, we performed cross-feeding experiments at different time points where we checked a co-culture of both species tracking the colony forming units and 16 S rRNA-based qPCR quantification of each species during growth (Figs. 5, 6). In this co-culture experiment, Baccell WH2 allowed *Bi. longum* subsp. *longum* NCIMB 8809 and *Bi. breve* UCC2003 to grow when both strains are cultivated with the intact fungal β-glucan as is obvious from viable count assessments (Fig. 5A for *Bi. longum* subsp. *longum* and 5 C for *Bi. breve*) and percentage of both bacteria in the co-culture (Fig. 5B for *Bi. longum* subsp. *longum* and 5D for *Bi. breve*). The observation of *Bifidobacterium* growth when in co-culture with Baccell WH2 in the presence of β-glucan agreed with the monoculture fermentation findings, when cell-free supernatant was used as carbon source, as displayed in Fig. 4.

Similarly, BT allowed growth of *Bi. breve* UCC2003 and *Bi. longum subsp. longum* NCIMB 8809 in co-culture as can be observed in Figs. 6A and 6C (colony forming units) and 6B and 6D (percentage), respectively. Again, it confirmed the ability of *Bacteroides* to allow for specific cross-feeding networks with *Bifidobacterium* strains in the gut when the former bacteria grow on dietary fungal β-glucan.

Finally, to expand this study with other commensal members of the human gut microbiota, we performed the co-culture experiment of Baccell WH2 and BT as primary and *Lactiplantibacillus plantarum* WCFS1 as a secondary degrader to confirm the ability of Baccel WH2 and BT to allow growth of this commensal. Figs. S2A and S2B for Baccell WH2 and Figs. S2C and S2D for BT showed this ability in co-culture too.

**Ability of *Bi. breve* UCC2003 to utilize β-1,6-glucobiose**. As stated above, *Bi. breve* UCC2003 can use glucobiose released by Baccell WH2 when grown on β-glucan. We conducted a bioinformatic analysis on the genome of the former bacterium to identify glycoside hydrolases that would allow *Bifidobacterium* to utilise this oligosaccharide. We identified a GH1 (Bbr_0109) which had previously been shown to act on cellobiose as substrate[53]. We hypothesised that this protein would act on gluco-oligosaccharides with different linkages as substrates. To prove our hypothesis, we recombinantly expressed the protein and performed enzymatic assays with β-1,4, β-1,3 and β-1,6-glucobiose as substrates of the enzyme. Bbr_0109 was active on β-1,4 and β-1,6-glucobiose as we expected, and this activity is shown by HPLC in Figs. S3A and S3B, respectively. In addition, we were able to characterise this activity and the kinetic parameters are calculated in Table 1. Bioinformatics analysis of the *Bi. longum subsp. longum* genome didn't reveal any homolog of Bbr_0109, or other candidate enzymes responsible for its ability to cross-feed on β-glucan-derived oligosaccharides and further work is therefore required to unravel the metabolic pathway responsible for this activity.

**Metabolites released by *Bacteroides/Bifidobacterium* co-cultivation on β-glucan**. After we confirmed the ability of *Bacteroides* to specifically allow *Bifidobacterium* growth when the

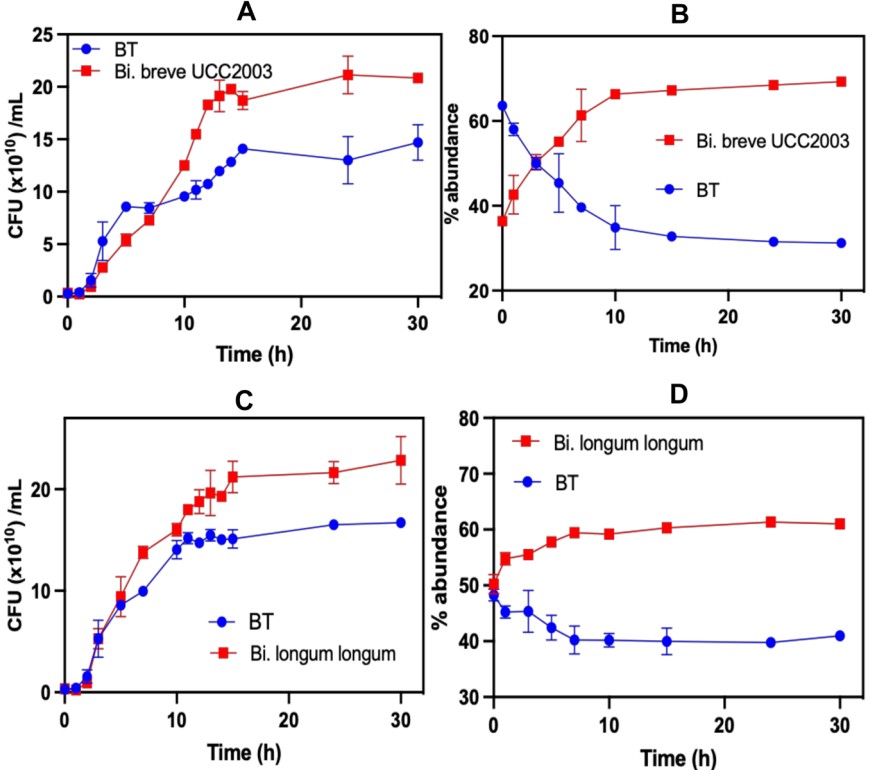

**Fig. 6 Cross-feeding experiments between BT and *Bifidobacterium* and *Lactiplantibacillus* spp. A** Colony forming units of BT + *Bi. breve* UCC2003. **B** Percentage of BT + *Bi. breve* UCC2003. **C** Colony forming units of BT + *Bi. longum* subsp. *longum*. **D** Percentage of BT + *Bi. longum* subsp. *longum*. All cross-feeding experiments have been produced in 3 different independent replicates (*n* = 3).

**Table 2 Millimolar (mM) concentration of the metabolites generated in the cell-free supernatant of MM + 1% β-glucan, following 24 h incubation with Baccell WH2 and *B. breve* UCC2003.**

|  | Succinate (mM) | Acetate (mM) | Lactate (mM) | Formate (mM) |
|---|---|---|---|---|
| No inoculum | ND | 0.9 ± 0.1 | 0.10 ± 0.01 | 1.8 ± 0.2 |
| Baccell WH2 | 78 ± 9.8 | 12 ± 2.2 | ND | ND |
| *B. breve* UCC2003 | ND | 1.1 ± 0.1 | 3.5 ± 0.4 | 2.6 ± 0.3 |
| Baccell WH2/*B. breve* UCC2003 | 99 ± 10.1 | 115 ± 16.2 | 21 ± 2.8 | 16 ± 1.9 |

Measures were made in triplicate.

former bacterium is cultivated on β-glucan, we performed a metabolomics analysis of the culture media to assess short chain fatty acid (SCFA)/lactate/succinate production by *Bacteroides* and *Bifidobacterium* in mono- and co-cultures. Table 2 lists the detected levels of acetate, butyrate, propionate, and lactate by these fermentations. Baccell WH2 was shown to produce succinate (78 mM) and acetate (12 mM) as main SCFAs produced in monoculture. As control, *Bi. longum* subsp. *longum* NCIMB 8809 failed to produce any significant amounts of SCFAs because of its inability to use the intact mycoprotein β-glucan. In contrast, in co-culture, acetate was the higher concentrated SCFA (115 mM) followed by succinate (99 mM) as consequence of the *subsp. longum* NCIMB 8809 growth. Finally, formate (16 mM) and lactate (21 mM) were also produced in co-culture because of the ability of *Bifidobacterium* to produce these metabolites.

## Discussion

β-glucan has previously been demonstrated to act as a prebiotic carbohydrate with various reported beneficial outcomes for human health[54–57]. Recently, Dejean et al. (2020) showed the ability of *Bacteroides* strains to utilise β-(1,3)-glucans, such as that derived from yeast or algal laminarin[13]. These authors showed that *Bacteroides uniformis* ATCC 8492 employs a specific polysaccharide utilisation locus for the deconstruction of these β-(1,3)-glucans. This genetic locus encodes a cell surface-associated glycoside hydrolase family 158 (GH158) enzyme and a GH16 to initiate depolymerization of the polysaccharide to release oligosaccharides that are incorporated by the SusC/D-like pair into the periplasm where a GH3 (β-glucosidase) continues with the degradative process converting all oligosaccharides into glucose monomers, which then enter central glycolytic catabolism. Also in 2020, Singh et al.[37] showed the ability of *Bacteroides uniformis* JCM 13288 to degrade β-(1,3)-glucans with a similar enzymatic mechanism to the afore mentioned *Bacteroides uniformis* strain. However, these latter authors revealed an additional GH30 activity able to degrade β-(1,6)-glucan side chains of the molecule releasing β-1,6-glucobiose. In addition, they demonstrated that the oligosaccharides released by this *Ba. uniformis* strain can be used by other members of the gut microbiota, which either did not grow or grew poorly on laminarin.

In the case of barley β-glucan, Tamura et al.[34–36] showed that Bacova and Baccell WH2 use a surface associated GH16 enzyme (Bovatus_03149 and BcellWH2_04354) to initiate the degradative process. In addition, the same authors showed that there are two GH3 enzymes present in the periplasm of Bacova (Bovatus_03146 and Bovatus_03153) and one in that of Baccell WH2 (BcellWH2_4356) to complete the metabolism of barley β-glucan.

In the work described here, we reveal an alternative pathway followed by Baccell WH2 to degrade dietary fungal β-glucans,

such as the polysaccharide derived from *Fusarium venenatum*, the main ingredient employed in Quorn® products, where the chemical structure consists of a linear β-(1,3)-glucan backbone with β-(1,6)-glucan as side chains (Fig. 1A). For this metabolism, Baccell WH2 employs two GULs to degrade this complex glycan, as revealed by proteomics and confirmed by transcriptomics (Fig. 1C and S1B, respectively). Within these GULs, Baccell WH2 encodes a novel GH157 and a GH30_3, both predicted to be located at the cell surface, to start the degradation of the complex glycan. These GULs were also shown to encode additional glycoside hydrolases required to fully degrade the gluco-oligosaccharides released by the outer membrane surface proteins and incorporated into the periplasm by the SusC/D-like pairs. These periplasmic proteins represent typical β-glucosidases belonging to GH families 3 and 2. Finally, this GUL encodes a protein with an unknown function (BcellWH2_02538), with a predicted location to be in the surface of Baccell WH2. This protein is in a perfect location to be a Surface Glycan Binding Protein (SGBP), which has been shown to help the SusC/D-like pair in the binding of oligosaccharides at the bacterial cell surface[35,36,58]. The reason for the presence of three different β-glucosidases (two GH3 and one GH2) encoded by these GULs to target fungal β-glucan remains speculative to us. We postulate that Baccell WH2 could act on several types of β-glucans, not only fungal sources, with different chemical structures. For that reason, they use distinct β-glucosidases to hydrolyse different linkages in these gluco-oligosaccharides. However, we can't rule out the possibility that some of these β-glucosidases are redundant and the bacterium is evolving to remove some of these genes from its genome under the high selection pressure imposed by the gut environment.

Finally, we show that Baccell WH2 and BT can share oligosaccharides with other members of the gut microbiota, including commensals *Bifidobacterium* and *Lactiplantibacillus species*. In this respect, Baccell WH2 was shown to allow cross-feeding interactions with *Bi. longum* subsp. *longum*, *Bi. breve* UCC2003 and *Lactiplantibacillus plantarum* WCFS1 (Fig. 5 and S2). In addition, BT was shown to promote specific cross-feeding with *Bi. longum* subsp. *longum*, *Bi. breve* UCC2003 and *Lactiplantibacillus plantarum* WCFS1 (Fig. 6 and S2), enabling growth of both bacteria in co-culture. Since the oligosaccharides released by BT and Baccell WH are different (i.e. glucoheptasaccharide and glucobiose), it will select for specific interactions in the gut. *Bi. breve* UCC2003 encodes a GH1 (Bbr_0109) to degrade the glucobiose released by BT. This GH1 is very specific for glucobiose, being active on β-(1,3)glucobiose, β-(1,4)glucobiose and β-(1,6) glucobiose (Table 1). However, the inability to act on glucotriose or other longer oligosaccharides may explain the partial ability of *Bi. breve* to use the heptasaccharide released by Baccell WH2 and, as consequence, its inability to grow in Baccell WH2 supernatant. These specific interactions in the cross-feeding between *Bacteroides* and *Bifidobacterium* members have been shown for other polysaccharides such as arabinogalactan[31], arabinoxylan[48] or inulin[59].

In terms of β-glucan, little has been reported on cross-feeding interactions by members of the gut microbiota. As we stated above, Singh et al.[37] showed the ability of *Ba. uniformis* JCM 13288 to share gluco-oligosaccharides with *Blautia producta* JCM1471, *Ruminococcus faecis* JCM15917 *and Bifidobacterium adolescentis* JCM 1275 when they grow on laminarin as the sole carbon source. In addition, Centanni et al.[60] showed interactions between *Bacteroides ovatus*, *Subdoligranulum variabile*, and *Hungatella hathewayi* using barley β-glucan as a carbon source. They showed that *Ba. ovatus* released 3-O-β-cellobiosyl-d-glucose and 3-O-β-cellotriosyl-d-glucose into the medium as final products and these oligomers enabled growth of the other two

bacteria with preference for 3-O-β-cellotriosyl-d-glucose in the case of *Hungatella hathewayi*. However, to the best of our knowledge our study for the first time reveals detailed molecular interactions between different members of *Bacteroides spp*. and other human commensals (*Bifidobacterium* and *Lactiplantibacillus*) when using fungal β-glucan.

Finally, we have shown the ability of Baccell and *B. breve* UCC2003 to produce different SCFA in mono- and coculture as a result of β-glucan fermentation. Baccell produces acetate and succinate as main metabolites when grow in monoculture on β-glucan, whereas co-cultivation with *B. breve* UCC2003 was shown to result in the production of not only acetate and succinate but also lactate and formate, the latter from *Bifidobacterium* fermentation. This production is in accordance with other fermentative processes where *Bacteroides* produces acetate as the main SCFA[31]. Munoz et al. have reported on the production of lower levels of succinate (60 mM) for AGP fermentation than for β-glucan (78 mM, Table 2) when Baccell was cultivated in mono-cultures. However, the opposite effect is observed for acetate production as Baccell was shown to produce 24 mM and 12 mM acetate when cultivated on AGP or β-glucan, respectively[31]. However, SCFA production appears to be higher for all metabolites when Baccell was co-cultivated with *B. breve* UCC2003 in the presence of β-glucan (115, 99, 21 and 16 mM for acetate, succinate, lactate and formate, respectively), when compared to co-cultivation in the presence of AGP (94, 68, 6 and 6.9 mM for acetate, succinate, lactate and formate, respectively)[31].

All data presented in this work allowed us to obtain a general model for the degradation of fungal β-glucan in Baccell WH2 and its interaction with bifidobacteria as secondary degraders (Fig. S3C).

In conclusion, this paper shows the ability of different members of human gut *Bacteroides* strains to use dietary fungal β-glucan. We have shown that *Bacteroides* employs two main GULs to degrade this complex glycan with novel enzymatic activities and families discovered in those GULs and that they share oligosaccharides and *Bi. longum* subsp. *longum* and *Bi. breve* UCC2003 selectively where able to use them. Finally, we have shown the specific enzyme (Bbr_0109, GH1) encoded in *Bi. breve* UCC2003 responsible for the degradation of the oligosaccharide released by BT. Thus, the study provides evidence that fungal β-glucan utilisation genes are present not only into *Bacteroides* but also into Gram-positive bacteria. Diverse β-glucans GULs identified in this study may pave the way for the development of engineered functional foods for the improvement of human health through proper nutritional intervention therapy.

## Methods

**Reagents**. Yeast and barley β-glucan tested in this study were purchased from Megazyme (Dublin, Ireland). D-Galactose and D-glucose were purchased from Sigma (United Kingdom). Luria-Bertani (LB) growth medium was purchased from Formedium (Norfolk, U.K.), reinforced clostridial agar from Oxoid Ltd. (Basingstoke, England) and Brain Heart Infusion from Sigma (United Kingdom). All reagents were of analytical grade.

**Mycoprotein β-glucan extraction and purification**. Cell wall material from *Fusarium venenatum* was extracted and the different glycan parts were purified according to a previously described method[61]. Briefly, cell wall material was alkaline extracted with 3% NaOH for 75 h obtaining two fractions: fraction 1 soluble in bases and fraction 2, insoluble in bases. Fraction 1 was shown to contain mannoproteins and β-1,6-1,3-glucan and fractions contained this β-1,6-1,3-glucan associated with chitin. Fraction 1 was neutralized with glacial acetic acid and centrifuged at 15,000 × g during 30 min and the supernatant subjected to a 24 h dialysis. After dialysis, the mixture was freeze-dried prior to its use (Fig. S4A, B).

**Growths, Proteomics and RT-qPCR**

*Growth*. Ba. ovatus ATCC8483 (Bovatus), Ba. thetaiotaomicron VPI-5482 (BT), Ba. cellulosilyticus DSMZ14838 (Baccell DSM) and Ba. cellulosilyticus WH2 (Baccell WH2) are capable of growth on all carbon sources of interest in this study and,

therefore, were cultured directly in 3 ml of Minimal Media (MM) containing 1% (wt/vol) carbohydrate, as described above[48].

*q-PCR*. *Bacteroides sp*. were precultured on MM + Glucose, pelleted, washed, resuspended twice in MM without any carbon source, and inoculated to an $A_{600}$ of ~0.3 in 4 ml of MM containing 1% (wt/vol) carbohydrate. Bacterial cultures were harvested in triplicate (at mid-log phase [$A_{600}$, ~0.6] after 5 h of incubation for Bovatus, BT, Baccell DSM and Baccell WH2, placed in RNA protect (Qiagen) for immediate stabilisation of RNA, and then stored at –20 °C. RNA was extracted and purified with the RNeasy minikit (Qiagen), and RNA purity was assessed by spectrophotometry. One µg of RNA was used for reverse transcription and synthesis of the cDNA (SuperScript VILO master mix; Invitrogen). Quantitative PCRs (20 µl final volume) using specific primers were performed with a SensiFast SYBR Lo-ROX kit (Bioline) on a 7500 Fast real-time PCR system (Applied Biosystems) (Table S2). Data were normalised to 16 S rRNA transcript levels, and changes in expression levels were calculated as fold change compared with levels for cultures of MM containing glucose.

*Proteomics*. Baccell WH2 was cultured in the same way as before with 1% β-glucan as complex polysaccharide and glucose as monosaccharide control. Bacterial cultures were harvested in triplicate after growth in MM+Glc or MM+β-glucan. Cells were collected by centrifugation (3,500 g, 15 min, 4 °C) and washed three times with PBS pH 7.4. Cell pellets were subsequently resuspended in 8 M urea buffer in 50 mM triethylammonium bicarbonate, containing 5 mM tris(2-carboxyethyl) phosphine. Cells were lysed via sonication using an ultrasonic homogenizer (Hielscher). Proteins were subsequently alkylated for 30 min at room temperature using 10 mM iodoacetamide in the dark. Protein concentration was determined using a Bradford protein assay (Thermo Fisher Scientific). Protein samples, containing 50 µg total protein, were diluted fivefold with 50 mM triethylammonium bicarbonate and protein digestion was performed at 37 °C for 18 h with shaking at 300 r.p.m. A protein to trypsin ratio of 50:1 was used. Trypsin digestion was stopped, and peptides were desalted as described above.

*Mass spectrometry*. Peptides were dissolved in 2% acetonitrile containing 0.1% trifluoroacetic acid, and each sample was independently analysed on an Orbitrap Fusion Lumos Tribrid mass spectrometer (Thermo Fisher Scientific), connected to an UltiMate 3000 RSLCnano System (Thermo Fisher Scientific). Peptides were injected on an Acclaim PepMap 100 C18 LC trap column (100 µm ID × 20 mm, 3 µm, 100 Å) followed by separation on an EASY-Spray nanoLC C18 column (75 µm µm × 500 mm, 2 µm, 100 Å) at a flow rate of 300 nlmin⁻¹. Solvent A was water containing 0.1% formic acid, and solvent B was 80% acetonitrile containing 0.1% formic acid. The gradient used for analysis of surface-shaved samples was as follows: solvent B was maintained at 3% for 6 min, followed by an increase from 3 to 35% B in 43 min, 35–90% B in 0.5 min, maintained at 90% B for 5.4 min, followed by a decrease to 3% in 0.1 min and equilibration at 3% for 10 min. The gradient used for analysis of proteome samples was as follows: solvent B was maintained at 3% for 6 min, followed by an increase from 3 to 35% B in 218 min, 35–90% B in 0.5 min, maintained at 90% B for 5 min, followed by a decrease to 3% in 0.5 min and equilibration at 3% for 10 min. The Orbitrap Fusion Lumos was operated in positive-ion data-dependent mode using a modified version of the recently described charge-ordered parallel ion analysis (CHOPIN) method for synchronized use of both the ion trap and the Orbitrap mass analysers. The CHOPIN method is derived from the 'Universal Method' developed by Thermo-Fisher, to extend the capabilities of mass analyser parallelization. The precursor ion scan (full scan) was performed in the Orbitrap in the range of 400–1600 *m/z* with a resolution of 120,000 at 200 *m/z*, an automatic gain control (AGC) target of 4 × 10⁵ and an ion injection time of 50 ms. MS/MS spectra of doubly charged precursor ions were acquired in the linear ion trap (IT) using rapid scan mode after collision-induced dissociation (CID) fragmentation. A CID collision energy of 32% was used, the AGC target was set to 2 × 10³ and a 300 ms injection time was allowed. Precursor ions with charge state 3–7 and with an intensity <5 × 10⁵ were also scheduled for analysis by CID/IT, as described above. Precursor ions with charge state 3–7 and with an intensity >5 × 10⁵ were, however, acquired in the Orbitrap (FT) with a resolution of 30,000 at 200 *m/z* after high-energy collisional dissociation (HCD). An HCD collision energy of 30% was used, the AGC target was set to 1 × 10⁴ and a 40 ms injection time was allowed. The number of MS/MS events between full scans was determined on-the-fly to maintain a 3 s fixed duty cycle. Dynamic exclusion of ions within a ± 10 p.p.m. *m/z* window was implemented using a 35 s exclusion duration. An electrospray voltage of 2.0 kV and capillary temperature of 275 °C, with no sheath and auxiliary gas flow, was used.

*Mass spectrometry data analysis*. All MS/MS spectra were analysed using Max-Quant 1.5.1.739 and searched against a database of *Bacteroides cellulosilyticus* MGS:158 (containing 4369 entries). Protein sequences were downloaded from Uniprot on 10 May 2020. Peak list generation was performed within MaxQuant and searches were performed using default parameters and the built-in Andromeda search engine. The enzyme specificity was set to consider fully tryptic peptides, and two missed cleavages were allowed. Oxidation of methionine, N-terminal acetylation and deamidation of asparagine and glutamine were allowed as variable modifications. A protein and peptide false discovery rate of less than 1% was

employed in MaxQuant. Proteins were confidently identified when they contained at least two unique tryptic peptides. Proteins that contained similar peptides and that could not be differentiated based on MS/MS analysis alone were grouped to satisfy the principles of parsimony. Reverse hits and contaminants were removed before downstream analysis. Skyline 4.1.0.11796 was used for extraction of ion chromatograms. Gene ontology enrichment was performed using PANTHER42 and subcellular protein localization prediction was performed using LocateP v243.

**Cloning, expression and purification of recombinant proteins**. The genes associated with the PULs described in Baccell WH2 [BccellWH2_01926 (GH3), BccellWH2_01931 (GH157), BaccellWH2_02537 (GH30_3)] and the proteins in BT [BT3312 (GH30_3) and BT3314 (GH3)] were amplified from Baccell WH2 and BT respectively using their genomic DNA as template and cloned into pET28a (Novagen) using NheI and XhoI restriction sites for production and purification of its encoded protein facilitated by the incorporation with an N-terminal His₆ tag (Table 2S). For this, *E. coli* TOP10 cells (ThermoFisher Scientific) were used. The recombinant construct was sequenced (Eurofins Genomics) to verify its genetic integrity and then used to transform *E. coli* BL21 (DE3) expression cells (Thermo-Fisher Scientific). Cells were cultured in Luria-Bertani (LB) medium containing 50 mg/ml kanamycin antibiotic to mid-log phase ($A_{600nm}$ of ~0.6). Protein expression was induced by adding 0.1 mM isopropyl β-D-1-thiogalactopyranoside (IPTG) to cells followed by growth overnight at 16 °C. The next day, cells were harvested by centrifugation (4000 × g) and re-suspended in the Talon buffer (20 mM Tris/HCl pH 8.0 plus 100 mM NaCl). Resuspended cells were disrupted and centrifuged (16,000 × g) for 20 min at 4 °C, after which recombinant proteins were purified from the resulting cell free extract by immobilised metal affinity chromatography (IMAC) using Talon™, a cobalt-based matrix. In the process, the Cell Free Extract (CFE) was loaded on a column containing the Talon resin and then washed with a Talon buffer. Another wash was performed with Talon buffer containing 10 mM imidazole followed by recombinant protein elution with 100 mM imidazole. Purified proteins were then exchanged into a buffer of choice by standard dialysis.

**Enzyme kinetics and product profile**. All enzyme assays, unless otherwise stated, were carried out in a 20 mM sodium phosphate buffer, pH 7.0, containing 150 mM NaCl and performed in triplicate. Assays were carried out at 37 °C employing 1 µM each GH [BccellWH2_01926 (GH3), BccellWH2_01931 (GH157), BccellWH2_02537 (GH30_3), BcellWH2_02541 (GH2), BT3312 (GH30_3) and BT3314 (GH3)] in the presence of 150 µM β-glucan. Aliquots were taken over a 16 h time course, and samples and products were assessed by TLC and high-pressure anion exchange chromatography (HPAEC) with pulsed amperometric detection (PAD). Sugars (mono and short oligosaccharides) were separated on a Carbopac PA1 guard and analytical column in an isocratic program of 100 mM sodium hydroxide for 40 min and then with a 100% linear gradient of 500 mM sodium acetate over 60 min. Sugars were detected using the carbohydrate standard quad waveform for electrochemical detection at a gold working electrode with an Ag/AgCl pH reference electrode.

In the case of GH30_3 and GH157, polysaccharide hydrolysis was quantified using a DNSA (dinitrosalicylic acid) reducing-sugar assay[48]. Assays were conducted in a final volume of 1 ml at the optimum pH and 37 °C for 10 min. Reactions were terminated by the addition of an equal volume (1 ml) of DNSA reagent. Colour was developed by heating to 80 °C for 20 min before reading $A_{540}$. Glucose (25 to 150 µM) was used to generate a standard curve for quantitation. To determine Michaelis-Menten parameters, different concentrations of polysaccharide solutions were used over the range of 0.025 to 3 mg ml⁻¹ with the appropriate concentration of enzyme for 10 min, and the numbers of reducing ends released were quantified as described above. The values were plotted using linear regression giving $k_{cat}/K_m$ as the slope of the line.

In the case of GH3s and GH2, the enzymatic assay was measured in the spectrophotometer at a wavelength of 340 nm measuring the releasing of glucose by the couple assay kit from Megazyme (Dublin, Ireland) for glucose release quantification.

**Cross-feeding experiments**. Before co-culture, Baccell WH2 and BT were grown in brain–heart infusion (BHI, Sigma Aldrich, UK) and washed twice in PBS. Monocultures of bifidobacterial strains were grown on Reinforced Clostridium Media (Oxoid, Basingstoke UK) and washed in PBS before being used to inoculate Minimal Media (MM) containing 1% Mycoprotein β-glucan[11]. Co-cultures were grown in inoculate Minimal Media containing 1% Mycoprotein β-glucan and the experiments were done in triplicate. Samples of 0.5 ml were taken at regular intervals during growth, which were serially diluted and plated onto BHI with agar and porcine haematin for determination of total CFUs per millilitre of the culture (Baccell WH2 and BT) and onto Reinforced Clostridium Media with 0.05% cysteine (for wild type bifidobacterial strains) and with 100 µg/ml erythromycin. *Lactiplantibacillus plantarum* WCSF4 was routinary grown on MRS media (Melford, U.K.) supplemented with vancomycin 10 µg/ml. Each Monoculture of *Bacteroides*, *Bifidobacterium* or *Lactiplantibacillus* was also plated for determination of CFUs per millilitre at intervals during cultivation.

Aliquots taken from mono and co-cultures were analysed by qPCR to quantify the ratio of each bacterial species during cultivation amplifying the 16 S rRNA gene

for each bacterium in the sample. We used specific primers (1 μM) for each bacterium for this qPCR and is shown in Table 2S.

Sugars (mixed type of oligosaccharides from growth media) were separated on a Carbopac PA200 guard and analytical column in an isocratic program of 100 mM sodium hydroxide for 40 min and then with a 100% linear gradient of 500 mM sodium acetate over a 60 min period. Sugars were detected using the carbohydrate standard quad waveform for electrochemical detection at a gold working electrode with an Ag/AgCl pH reference electrode.

**Structural prediction with AlphaFold and Phyre2.** 3-D structure of GH157 (BcellWH2_01931) was modelled using AlphaFold2 colab software We thank the AlphaFold team for developing an excellent model and open sourcing the software https://colab.research.google.com/github/sokrypton/ColabFold/blob/main/AlphaFold2.ipynb#scrollTo=UGUBLzB3C6WN). Bccell WH2_02537 (GH30_3) model was obtained using Phyre2 server[52]. Structures were visualised using Pymol (The PyMOL Molecular Graphic system, version 2.0 Schrodinger, LLC).

**Metabolite analysis by high performance liquid chromatography (GC/MS).** HPLC analysis was used to assess SCFA production by (cross-feeding) Baccell WH2 and Bi. Breve UCC2003. Growth medium supernatants from stationary phase (co-cultures) were sterilized (0.45 μM filter, Costart Spin-X column) and injected into an UltiMate® BioRS Thermo HPLC system (Thermo Fisher Scientific) with a refractive index detector system. This system was used to identify and calculate the production of acetate, lactate and butyrate as a result of carbohydrate fermentation. Concentrations were calculated based on known standards. Non-fermented medium containing LW-AG served as control. An Accucore™ C18 HPLC column was used and maintained at 65 °C. Elution was performed for 25 min using 10 mM $H_2SO_4$ solution at a constant flow rate of 0.6 ml min$^{-1}$.

**LC/MS of the oligosaccharides released by Baccell WH2 and BT.** The sample containing the oligosaccharides generated from the supernatant from Baccell WH2 and BT grown on β-glucan was diluted 1:10 (v/v) with Buffer B (85% acetonitrile/15% 50 mM ammonium formate in water, pH 4.7) and 0.5 μl was analysed by liquid chromatography–mass spectrometry analysis via elution from a ZIC-HILIC (SeQuant, 3.5 μm, 200 Å, 150 × 0.3 mm, Merck) capillary column. The column was connected to a NanoAcquity HPLC system (Waters) and heated to 35 °C with an elution gradient as follows: 100% Buffer B for 5 min, followed by a gradient to 25% Buffer B/75% Buffer A (50 mM ammonium formate in water, pH 4.7) over 40 min. The flow rate was 50 μl min−1 and 10 column volumes of Buffer B equilibration was performed between injections. MS data were collected using a Bruker Impact II QT of mass spectrometer operated in positive ion mode, 50–2000 m/z, with capillary voltage and temperature settings of 2800 V and 200 °C respectively, together with a drying gas flow and nebulizer pressure of 6 l min$^{-1}$ and 0.4 bar. The MS data were analysed using Compass DataAnalysis software (Bruker).

**Reporting summary.** Further information on research design is available in the Nature Portfolio Reporting Summary linked to this article.

## Data availability

All proteomic data are deposited in PeptideAtlas (identifier PASS04831, raw data for blank, glucose and mycoprotein β-glucan as carbon sources). All data generated during this study are included in this published article (and its supplementary information files). All source data underlying the graphs and charts can be found in Supplementary Data 1. Original SDS gels are available in Fig. S5. Further raw data is available upon request.

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

## Acknowledgements

J.M.-M. is supported via a PhD studentship partly funded by Marlow Foods Ltd. This work has been conducted independently of Marlow foods and the authors have no financial or other vested interest in the outcome of the work. J.M.-M. received financial support from an internal grant from Northumbria University. D.v.-S. is member of the APC Microbiome Ireland which receives financial support from Science Foundation Ireland (SFI/12/RC/2273 − P1 and SFI/12/RC/2273 − P2) as part of the Irish Government's National Development Plan.

## Author contributions

P.F.-J. cloned, expressed, and purified recombinant enzymes; conducted and analysed kinetics for hydrolysis of polysaccharides, oligosaccharides, and chromogenic substrates; determined hydrolysis products in the HPLC; performed growth curves of all anaerobic bacteria; carried out cross-feeding experiments; and wrote the article. W.C. conducted and analysed the proteomics data and wrote the proteomics section in the manuscript. G.B., D.V.S. and J.M.-M. designed and directed research and co-wrote the article with input from all authors.

## Competing interests

The authors declare no competing interests.
