## [Peer Review File · Communications Biology]

Reviewers' comments:

Reviewer #1 (Remarks to the Author):

COMMSBIO-22-4234-T

Title: Fungal β -glucan-facilitated cross-feeding activities between *Bacteroides* and *Bifidobacterium* species

-General remarks:

β -glucan has been established as a dietary polysaccharide supporting growth of particular gut-associated bacteria. This study explored the mechanism of β -glucan metabolism by intestinal commensal bacteria and verified the potential cross-feeding, which is an interesting research. However, there are several comments need to be addressed to improve quality of the manuscript.

-Specific remarks

- 1.SCFAs were mentioned in both methods and results of the co-culture system, but not in the discussion. No need to discuss the role of SCFAs in cross feeding? Please analyze.
- 2.Changes in GH enzymes during cross-feeding activities were described in this manuscript. Have you considered changes in the bifidobacterium transport system? If not, please explain why.
- 3.There should be more descriptions of the related research on Fungal β -glucan degradation in the Introduction part.
- 4.Line 40: replace "syntrophy" with "cross-feeding".
- 5.Line 108: "there are no...by Bacteroidota", No study? Confirm.
- 6.Line 115: The "AGP" should be in full name when it was mentioned in the first time.
- 7.Line 136: There are 23 *Bacteroides* species in the Table S1 instead of 20.
- 8.Line 142: Table S1 shows that *Ba. vulgatus* was unable to utilize yeast β -glucan, please check.
- 9.Line 775: There are four *Bacteroides* species shown in Fig. 1B.
- 10.The authors should check the manuscript to avoid the above mistakes.

Reviewer #2 (Remarks to the Author):

"Fungal β -glucan-facilitated cross-feeding activities between *Bacteroides* and *Bifidobacterium* species" by Pedro Fernandez-Julia et al. discuss an interesting topic i.e. the mechanism of cross-feeding between members of the two key gut microbial genera *Bacteroides* and *Bifidobacterium*. In this context, the authors focus on a β -glucan of fungal origin.

The manuscript is well written with a clear English. Moreover, the manuscript provides valuable information regarding the complex topic of microbe-microbe metabolic interactions.

My only comments are:

- 1) The introduction is quite long, thus I think the description of the *Bacteroides* and *Bifidobacterium* genera can be condensed.
- 2) Is not clear to me if the ability of bifidobacterial strains *Bi. Breve* UCC2003, *Bi. longum* subsp. *longum* NCIMB 8809 and *Lb. plantarum* WCFS1 to use heptasaccharide released by *Bacell* WH2 represent a strain-specific feature or an assumed species-wide feature. The authors tested other strains for these species?
- 3) Genes homologous to those identified as participating in the cross-feeding effect also present in other strain of the species or in other species of the same genus? This may help to understand the prevalence of this phenomenon/mechanism.

Reviewer #3 (Remarks to the Author):

The authors found syntrophic interactions between *Bacteroides* species and several probiotic species on β -glucan obtained from *Fusarium venenatum*. As little is known about how this commercialized mycoprotein β -glucan is degraded in the human gut microbial community members, this study could trigger related research and thus potentially has a significant impact in the related fields. However, the manuscript contains a lot of concerns regarding methodology and

interpretation that should be addressed before publication.

Line 69: It is appropriate to cite Terrapon's PUL database paper (PMID: 29088389) here, in addition to CAZy database.

Line 80: It is appropriate to cite the paper by Sakanaka et al. (PMID: 31489370) as that is the first to show that HMOs serve as a natural prebiotic in vivo.

Lines 104-107: Cite appropriate papers describing the structural difference in glucan (side)chains among the substrates.

Lines 134-136: The authors need to show the content of b-glucan in the purified preparation. According to Figure S4 and Supplemental Experimental Procedures, the supernatant obtained after neutralization with acetic acid contains mannoproteins and b-glucan, however, the HPAEC-PAD chromatogram shows the Glc peak only. How did the authors separate mannoproteins from mycoprotein b-glucan. And, how much Glc is contained in the preparation. The authors can estimate the b-glucan abundance (weight percent) in the preparation by measuring the Glc concentration of TFA-treated sample. Determination of the b-glucan content in the preparation is indispensable for evaluating k_{cat}/K_m values among different substrates.

Line 168: Please define HTCS (hybrid two component system?)

Line 195: Has it already known that BaccellWH2 strain utilizes barley glucan? If so, please cite an appropriate paper.

Line 202: The authors are requested to show the results of SDS-PAGE of the purified enzymes in the supplemental information file, because they determined the kinetic parameters of them.

Linea 213-214: The referee was unable to find the chromatogram obtained for pustulan incubated with GH157.

Line 259: The authors did not determine the structure. They only presented the structure model.

Lines 263-264: The authors mentioned here that they built a structure model of GH30_3 (WH2_02537) by comparing it with BT3312, but in the method section, they mentioned that they used alphafold-2. Which algorithm, swiss-model or alphafold2, did the authors use for predicting the structure?

Line 269: Not Fig. 4A, but Table 1.

Line 287: Not Fig, 3D, but Fig. 3E.

Lines 287-289: What adduct of hepta-glucose gives m/z of 1197? The exact m_s of hepta-glucose is 1152.38. The m/z of the highest peak seems to be around 1170 in Fig 3E (not 1197).

Line 293: Not Fig. 3C but Fig. 3D.

Lines 314-318. The referee sees that OD600 values reaches 0.8 for *B. longum* and 0.6 for other two strains in Fig. 4A.

Line 330: Which primer sets did the authors use for quantifying three bacterial strains in the co-culture experiments? The referee was unable to find a description in the method section.

Line 397: gentiobiose. As the authors use the terms "glucobiose and glucotriose" throughout the text, gentiobiose could be better to be replaced with b1,6-glucobiose.

Line 503: For what purpose did the author use galactose in their experiments?

Line 517: Please specify components of the minimal medium or cite an appropriate paper.

Figure 4B: Not infantis, but longum.

Table 1: Please clearly describe the lack of activity of enzymes towards the tested substrates by mentioning with Not detectable, trace (< 0.1), etc. The linkage preference is one of the most important characteristics of GHs.

Reviewer #1 (Remarks to the Author):

COMMSBIO-22-4234-T

Title: Fungal β -glucan-facilitated cross-feeding activities between *Bacteroides* and *Bifidobacterium* species

-General remarks:

β -glucan has been established as a dietary polysaccharide supporting growth of particular gut-associated bacteria. This study explored the mechanism of β -glucan metabolism by intestinal commensal bacteria and verified the potential cross-feeding, which is an interesting research. However, there are several comments need to be addressed to improve quality of the manuscript.

-Specific remarks

1. SCFAs were mentioned in both methods and results of the co-culture system, but not in the discussion. No need to discuss the role of SCFAs in cross feeding? Please analyze.

Answer: As the reviewer stated, SCFA production has been discussed in the revised version of the paper and compared with other fibre fermentations.

2. Changes in GH enzymes during cross-feeding activities were described in this manuscript. Have you considered changes in the bifidobacterium transport system? If not, please explain why.

Answer: We have considered the changes in GH enzymes during cross-feeding activities in the *Bifidobacterium* transport system. We performed a similar approach with proteomics to understand the proteins expressed in that bifidobacterial system. However, we didn't obtain a clear pathway expressed in the experiments. Because of that, we will embark on an RNAseq analysis, but this will be beyond the scope of the current paper.

3. There should be more descriptions of the related research on Fungal β -glucan degradation in the Introduction part.

Answer: We have expanded the research background on fungal β -glucan metabolism in the introduction part, as requested by the Reviewer.

4. Line 40: replace "syntrophy" with "cross-feeding".

Answer: We have changed the word as requested by the Reviewer.

5. Line 108: "there are no...by Bacteroidota", No study? Confirm.

Answer: The reviewer is correct, only a few papers are showing a benefit on human health. We have amended the sentence and included the relevant papers as references.

6. Line 115: The "AGP" should be in full name when it was mentioned in the first time.

Answer: The full name has been added as requested by the Reviewer.

7.Line 136: There are 23 *Bacteroides* species in Table S1 instead of 20.

Answer: The Reviewer was absolutely right, we have amended the number to 23.

8.Line 142: Table S1 shows that *Ba. vulgatus* was unable to utilize yeast β -glucan, please check.

Answer: The Reviewer is correct, we have corrected this mistake in the relevant Table.

9.Line 775: There are four *Bacteroides* species shown in Fig. 1B.

Answer: The Reviewer is correct, we have amended this oversight in the revised version of the manuscript.

10.The authors should check the manuscript to avoid the above mistakes.

Answer: We have checked the entire manuscript to avoid the mistakes correctly pointed by the Reviewer.

Reviewer #2 (Remarks to the Author):

"Fungal β -glucan-facilitated cross-feeding activities between *Bacteroides* and *Bifidobacterium* species" by Pedro Fernandez-Julia et al. discuss an interesting topic i.e. the mechanism of cross-feeding between members of the two key gut microbial genera *Bacteroides* and *Bifidobacterium*. In this context, the authors focus on a β -glucan of fungal origin.

The manuscript is well written with a clear English. Moreover, the manuscript provides valuable information regarding the complex topic of microbe-microbe metabolic interactions.

My only comments are:

1) The introduction is quite long, thus I think the description of the *Bacteroides* and *Bifidobacterium* genera can be condensed.

Answer: We have reduced the description for *Bacteroides* and *Bifidobacterium* in the introduction section. In addition, we have removed some references to be more concise. However, one of the other Reviewers requested to incorporate more evidence of fungal β -glucan metabolism in the introduction part, so the length of this section, at the end, is quite like the original version but with some modifications to satisfy all Reviewers' requests.

2) Is not clear to me if the ability of bifidobacterial strains *Bi. Breve* UCC2003, *Bi. longum* subsp. *longum* NCIMB 8809 and *Lb. plantarum* WCFS1 to use heptasaccharide released by *Baccell* WH2 represent a strain-specific feature or

an assumed species-wide feature. The authors tested other strains for these species?.

Answer: Yes, we have tested several other Bifidobacterium and Lactiplantibacillus strains but, for clarity, we only incorporated data obtained for *Bi. breve* UCC2003, *Bi. longum* subsp. *longum* NCIMB 8809 and *Lb. plantarum* WCFS1 in this paper. In this case, we have tested one or more strains of the following species: *Bi. longum* subsp. *infantis*, *Bi. longum* subsp. *longum*, *Bi. adolescentis*, *Bi. breve*, *Bi. bifidum*, *Lb. paraplantarum* or *Lb. pentosus*.

3) Genes homologous to those identified as participating in the cross-feeding effect also present in other strain of the species or in other species of the same genus? This may help to understand the prevalence of this phenomenon/mechanism.

Answer: Interestingly, the gene encoding the GH1 enzyme in *Bi. breve* UCC2003 was only found in certain strains of this species according to the Blastp search we did against Bifidobacterium genera. However, it is still possible that other genes, distanced related to the GH1 found in *Bi. breve* UCC2003, would be active on the oligosaccharides released by Bacteroides sp. We would need to perform a systematic generic carbon screen with these bacteria and see which specific strain is able to degrade these oligosaccharides to, after that, perform RNA-seq on those strains to dissect the genes responsible of that metabolism.

Reviewer #3 (Remarks to the Author):

The authors found syntrophic interactions between Bacteroides species and several probiotic species on b-glucan obtained from Fusarium venenatum. As little is known about how this commercialized mycoprotein b-glucan is degraded in the human gut microbial community members, this study could trigger related research and thus potentially has a significant impact in the related fields. However, the manuscript contains a lot of concerns regarding methodology and interpretation that should be addressed before publication.

Line 69: It is appropriate to cite Terrapon's PUL database paper (PMID: 29088389) here, in addition to CAZy database.

Answer: Citation has been added as requested by the Reviewer.

Line 80: It is appropriate to cite the paper by Sakanaka et al. (PMID: 31489370) as that is the first to show that HMOs serve as a natural prebiotic in vivo.

Answer: Citation has been added as requested by the Reviewer.

Lines 104-107: Cite appropriate papers describing the structural difference in glucan (side)chains among the substrates.

Answer: We have included the references as pointed by the Reviewer.

Lines 134-136: The authors need to show the content of b-glucan in the purified preparation. According to Figure S4 and Supplemental Experimental Procedures, the supernatant obtained after neutralization with acetic acid contains mannoproteins and b-glucan, however, the HPAEC-PAD chromatogram shows the Glc peak only. How did the authors separate mannoproteins from mycoprotein b-glucan. And, how much Glc is contained in the preparation. The authors can estimate the b-glucan abundance (weight percent) in the preparation by measuring the Glc concentration of TFA-treated sample. Determination of the b-glucan content in the preparation is indispensable for evaluating k_{cat}/K_m values among different substrates.

Answer: The Reviewer has a valid point as we didn't specify how to separate mannoproteins from b-glucan. Essentially, we did an ethanol precipitation as outlined in reference 61. In the revised version of the manuscript, we have modified figure S4 to include that step. After ethanol precipitation, we performed TFA hydrolysis and assessed the sample by HPLC which showed that we only had b-glucan as purified polysaccharide. Admittedly, we show the sample with only glucose as we had other samples where the purification was incomplete, with some remaining mannoproteins in the sample (in this case we saw glucose and mannose peaks in the HPLC chromatogram confirming the presence of both polysaccharides).

However, we didn't estimate the concentration for b-glucan in Molar as we wanted to compare our results with other authors where they used the kinetic parameters in mg/ml/min (references 12 and 16 for b-1,3-glucan and pustulan respectively). Because of that we kept the results with those units as well.

Line 168: Please define HTCS (hybrid two component system?).

Answer: As the Reviewer requested, we have defined the full name for HTCS in the text.

Line 195: Has it already known that BaccellWH2 strain utilizes barley glucan? If so, please cite an appropriate paper.

Answer: As the Reviewer requested, the citation has been added where the authors showed growth and some transcriptomics on specific SusC/D with barley b-glucan as carbon source.

Line 202: The authors are requested to show the results of SDS-PAGE of the purified enzymes in the supplemental information file, because they determined the kinetic parameters of them.

Answer: As the Reviewer requested, we have incorporated a new Fig. in Supplementary Information (Fig. S5) with all SDS-gels for the recombinant proteins expressed in the study.

Linea 213-214: The referee was unable to find the chromatogram obtained for pustulan incubated with GH157.

Answer: We did not included this data for simplicity as this enzyme did not appear to be active on this substrate. In the revised version we have specified that this data is not shown in the text. However, we have included in Table 1 that GH157 was inactive on pustulan and that GH30_3 was also inactive on Linear β -(1,3)glucan.

Line 259: The authors did not determine the structure. They only presented the structure model.

Answer: The Reviewer is absolutely right so we have included modelling in the title of the subsection.

Lines 263-264: The authors mentioned here that they built a structure model of GH30_3 (WH2_02537) by comparing it with BT3312, but in the method section, they mentioned that they used alphafold-2. Which algorithm, swiss-model or alphafold2, did the authors use for predicting the structure?

Answer: The Reviewer was right and now we have amended the method section to clarify it and in the main document as well.

Line 269: Not Fig. 4A, but Table 1.

Answer: Table 1 has been added instead as correctly cited by the Reviewer.

Line 287: Not Fig, 3D, but Fig. 3E.

Answer: The Figure has been corrected.

Lines 287-289: What adduct of hepta-glucose gives m/z of 1197? The exact ms of hepta-glucose is 1152.38. The m/z of the highest peak seems to be around 1170 in Fig 3E (not 1197).

Answer: As the Reviewer pointed, the assigned peak was 1171 but, my mistake, we assigned 1197. The exact adduct is the heptasaccharide with NH₄ as adduct (extra 18 mass). In the revised version of the manuscript, we have corrected and specified this in the text.

Line 293: Not Fig. 3C but Fig. 3D.

Answer: The Figure has been corrected.

Lines 314-318. The referee sees that OD600 values reaches 0.8 for *B. longum* and 0.6 for other two strains in Fig. 4A.

Answer: We had a mistake reporting the final OD and, in the revised version, we have reported the correct final OD for all *Bifidobacterium* strains, as correctly reported by the Reviewer.

Line 330: Which primer sets did the authors use for quantifying three bacterial strains in the co-culture experiments? The referee was unable to find a description in the method section.

Answer: The Reviewer pointed correctly that we didn't include the primers used for the bacterium quantification. Now, we have incorporated those primers into Table 2S.

Line 397: gentiobiose. As the authors use the terms "glucobiose and glucotriose" throughout the text, gentiobiose could be better to be replaced with b1,6-glucobiose.

Answer: Gentibiose has been replaced by b1,6-glucobiose as requested by the Reviewer.

Line 503: For what purpose did the author use galactose in their experiments?

Answer: We didn't use it in the experiments, we only used as standard for the HPLC chromatograms. (Fig. 2).

Line 517: Please specify components of the minimal medium or cite an appropriate paper.

Answer: We cited the appropriate paper in the revised version of the paper.

Figure 4B: Not infantis, but longum.

Answer: We have corrected the figure as requested by the Reviewer.

Table 1: Please clearly describe the lack of activity of enzymes towards the tested substrates by mentioning with Not detectable, trace (< 0.1), etc. The linkage preference is one of the most important characteristics of GHs.

Answer: As requested by the Reviewer, we have incorporated the activities where the proteins did not appear to act upon certain substrates in Table 1.

REVIEWERS' COMMENTS:

Reviewer #1 (Remarks to the Author):

All our questions have been well answered.

Reviewer #2 (Remarks to the Author):

Dear Authors,

thanks for your efforts.

The overall quality of the manuscript has been drastically improved and now it meets the quality standards for publication.

Reviewer #3 (Remarks to the Author):

I appreciate that the authors have addressed the raised concerns. But, I still have a few comments, which I think can be fixed during the production process.

P12 L9: Fig. S1F is Fig. 2E?

P17 bottom: "co-culture" might be "Bacell WH supernatant", if I understand the results correctly.

P19: Supplemental information has now five figures.

P21 L4: I could not find the peak corresponding to Gal in Figure 2, though in the response letter, the authors say that Gal is used as a standard.

P22 L5: I could not find the results of enzyme assays using chromogenic substrates.

Fig. 2 and 3 legends: They lack descriptions about some panels.

Reviewer #3 (Remarks to the Author):

I appreciate that the authors have addressed the raised concerns. But, I still have a few comments, which I think can be fixed during the production process.

- P12 L9: Fig. S1F is Fig. 2E?.
 - We have corrected that Figure as correctly pointed by the Reivewer.
- P17 bottom: “co-culture” might be “Bacell WH supernatant”, if I understand the results correctly.
 - The Reviewer correctly understood the results so we have changed the sentence as pointed out.
- P19: Supplemental information has now five figures.
 - We have corrected that sentence as pointed by the Reviewer.
- P21 L4: I could not find the peak corresponding to Gal in Figure 2, though in the response letter, the authors say that Gal is used as a standard.
 - We have used galactose as monomeric standart. In Figure 2, in all HPLC traces we showed standards with different linkage (1,3 or 1,6-glucooligosaccharides). In those standards, G1 means galactose. There is where we used this monosaccharide.
- P22 L5: I could not find the results of enzyme assays using chromogenic substrates.
 - We have removed the sentence about chromogenic substrates as we only used them to check the stability of some exo-glycosidases.
- Fig. 2 and 3 legends: They lack descriptions about some panels.
 - Descriptions have been made for all panels in both figures as correctly pointed by the Reviewer.